# Fracture Toughness and Fracture Surface Morphology of Concretes Modified with Selected Additives of Pozzolanic Properties

**Janusz Konkol** 

Department of Materials Engineering and Technology of Building, Rzeszow University of Technology, PL-35959 Rzeszow, Poland; janusz.konkol@prz.edu.pl

**Abstract:** Modern methods of designing and testing concrete must be extended to appropriate material engineering approaches. It is then crucial to link the properties of concrete with its structure described in a quantitative way. The aim of the article was to present the results of research on concretes modified with three additives: Silica fume (SF), activated fluidal ash (FA), and metakaolinite (MK). The concretes were tested for compressive strength, fracture toughness (determining critical stress intensity factor $K_{Ic}{}^S$ and elastic modulus $E$). Also, stereological and fractal tests were performed. The research program covered three separate experiment plans, adopting the water/binder ratio and the additive/binder mass ratio as the independent variables. The results of experiments and their analysis proved a statistically significant relationship between fracture morphology (fractal dimension $D$) and concrete composition and fracture toughness. A higher fractal dimension was found in concretes with a higher content of cement paste and a lower content of additive. No significant effect of the type of additive used in the above dependence was found. An original method enabling the determination of mechanical properties of concrete with no need for destructive testing has been developed.

**Keywords:** fracture toughness; fractal dimension; stereology; concrete; pozzolanic additives; silica fume; activated fluidal ash; metakaolinite

---

## 1. Introduction

The specific nature of tests on fracture toughness makes them non-typical and used infrequently compared with the common experiments on compressive strength. However, the significance of fracture toughness tests is indisputable. The state of stress in concrete elements is not restricted merely to compression. An example of how stresses change in concrete is illustrated by the result of tests done by Dantu [1]. It turns out that in plain concrete, the stresses occurring at the contact between aggregate grain and cement matrix can be twice higher than the average stresses adopted in the design. The variable stress pattern, together with discontinuities in the concrete structure, proves the necessity of research on fracture toughness.

What is the measurable effect of the fracture, a result of stresses, is the resulting fracture surface or a profile line, separated out of it. The shape of the profile line is a resultant of concrete strength and the content of concrete components, dimension and number of defects existing in concrete (cracks, discontinuities on the aggregate/cement paste contact) quality and strength of aggregate/cement paste composition and concrete porosity.

Cracking is nowadays widely recognized as a fractal phenomenon, so fractal dimension can be employed to represent the geometric-statistical complexity of the fracture surface morphology. Studies were stimulated by the leading paper of Mandelbrot [2] on the fractal concept. Such studies proved the

usefulness of this concept for surface morphology in various material technologies, including that of the cementitious materials. Relationships could be derived between the fractal dimension *D*, on the one side, and the compressive strength $f_c$ [3–5], the fracture toughness as expressed by the critical stress intensity factor $K_c$ or the fracture energy $G_F$ [4,6–15] and the elastic modulus *E* [4], on the other side.

Owing to fractal geometry, the shape of the fracture surface or a profile line can be described with a single parameter, i.e., fractal dimension [3–24], which depends on the extent of particular phases in the concrete. The fracture surface can also be described with other fractographic parameters [25–32].

Quantitative analyses of concrete composition have been supplemented with the results of stereological tests performed on plane sections.

To obtain statistical relations the results of fractal and stereological tests have been referred to the properties of additive modified concretes.

The author's research concerning one of the applied additives (meakaolinite) showed the possibility of applying this approach [32]. In this paper, the research has been extended to the analysis of concretes with the addition of silica fume and activated fluidal ash.

## 2. Materials and Methods

Tests were performed for three additives separately: Silica fume (SF), activated fluidal ash (FA), and metakaolinite (MK). Metakaolinite and fluidal ash as additives for concrete have been and still are the subject of numerous studies [32–53]. The additives were used as a partial substitute for cement. The theory of experiment planning was applied to obtain an optimum experiment program. A central composition plan was adopted according to which nine different recipes of concrete mix were made, with the experiment repeated in the central point. The mix composition differed in variable water/binder ratio (*w/b*) and the variable content of additive relative to the binder mass. The water/binder ratio ranged from 0.35 to 0.54, while the additive content from 2.1 to 14.9%, relative to the binder mass, i.e., from 2.2 to 17.5% of cement mass in case of activated fluidal ash and metakaolonite, and from 1.8 to 10.2% relative to binder mass, i.e., from 1.8 to 11.4% of cement mass in the case of silica fume.

The following components were used: Portland cement CEM I 32.5R, quartz sand to 2 mm with specific gravity of 2.65 kg/dm$^3$, basalt grit to 16 mm with specific gravity of 3.06 kg/dm$^3$ and alternately: Silica fume (with 94% $SiO_2$), activated fluidal ash (with 40 % $SiO_2$, 30% $Al_2O_3$ and 13% CaO content) or metakaolinite (with 53% $SiO_2$ and 42% $Al_2O_3$ content). The silica fume SikaFume$^®$ HR used is characterized by specific gravity of 2.2 kg/dm$^3$ and a specific surface of 22 m$^2$/g. The fluidal ash used was a mechanically activated fluidal ash. Compared with non-activated fluid ashes, activated fluidal ash accelerates the setting processes. This results from deagglomeration and structural defects on the surface of the ash particles. The specific gravity of activated fluidal ash is 2.53 kg/dm$^3$, and the specific surface is 0.51 m$^2$/g. Metakaolin is a pozzolanic material. It is obtained by the calcination of kaolinitic clay at a temperature ranging between 550 °C and 800 °C. The metakaolinite used in the test was obtained in the process of kaolin calcination at a temperature of around 800 °C, and its specific gravity was 2.54 kg/dm$^3$. A diversity of grain sizes was observed, ranging from 0.1 µm to 100 µm, with the majority of sizes in the range from 1 µm to 10 µm (constituting around 60%). The share of grains below 1 µm was around 20% and the share of grains below 17 µm was 90%.

The concrete mix composition was determined within the experimental plan (see Table 1). The consistency tests performed on all the concrete mixes by a flow table test produced a flow of 0.41 ± 0.03 m. The concrete mix constant consistency was obtained using superplasticizer FM-6 in the amount determined by the laboratory tests.

**Table 1.** Concrete mix components and type of additive.

| Series No. | Variable | | Concrete Mix Composition in kg After the Adopted Plan | | | | | |
| | *w/b* | FA/*b* or MK/*b* (SF/*b*) [1] | Binder | Cement [1] | FA or MK (SF) [1] | Water | Sand | Basalt |
|---|---|---|---|---|---|---|---|---|
| 1 | 0.380 | 0.04 (0.03) | | 435.8 (440.4) | 18.2 (13.6) | 172.5 | | |
| 2 | 0.380 | 0.13 (0.09) | | 395.0 (413.1) | 59.0 (40.9) | 172.5 | | |
| 3 | 0.510 | 0.04 (0.03) | | 435.8 (440.4) | 18.2 13.6) | 231.5 | | |
| 4 | 0.510 | 0.13 (0.09) | | 395.0 (413.1) | 59.0 (40.9) | 231.5 | | |
| 5 | 0.353 | 0.085 (0.06) | 454 | 415.4 (426.8) | 38.6 (27.2) | 160.3 | 739.3 | 1212.5 |
| 6 | 0.537 | 0.085 (0.06) | | 415.4 (426.8) | 38.6 (27.2) | 243.8 | | |
| 7 | 0.445 | 0.02 (0.02) | | 444.3 (446.0) | 9.7 (8.0) | 202.0 | | |
| 8 | 0.445 | 0.15 (0.10) | | 386.5 (407.5) | 67.5 (46.5) | 202.0 | | |
| 9, 10 | 0.445 | 0.085 (0.06) | | 415.4 (426.8) | 38.6 (27.2) | 202.0 | | |

[1] The values in brackets refer to concrete mixes with silica fume (SF).

The strength tests were performed after 28 days of curing of the specimens, stored in air-humidity conditions (air relative humidity >95%). The tests covered the determination of the compressive strength $f_c$, critical stress intensity factor $K_{Ic}{}^S$, and elastic modulus $E$.

For the compressive tests, cubes of sides of 0.1 m were used. For the fc compression test, 209 cubic test specimens were used, including 72 specimens of fluidal ash-modified concrete, 70 specimens of metakaolinite-modified concrete, and 67 specimens of concrete with silica fume addition.

The fracture toughness tests were carried out according to the I model (bending tension) on beams of dimensions $0.08 \times 0.15 \times 0.70$ m with an initial crack. The critical stress intensity factor $K_{Ic}{}^S$ and modulus $E$ were calculated after [54]. During the fracture toughness tests, the dependence of the loading force on the crack mouth opening displacement (CMOD) was recorded. Examples of plots of the load vs. CMOD obtained for concrete with fluidal ash addition are shown in Figure 1.

The critical stress intensity factor $K_{Ic}{}^S$ was calculated from the following relationship (Equation (1)):

$$K_{Ic}^S = 3\left(P_{max} + 0.5W\frac{S(\pi a_c)^{0.5}F(\alpha_2)}{2d^2b}\right), \text{ N/m}^{3/2} \tag{1}$$

in which:

$$F(\alpha_2) = \frac{1.99 - \alpha_2(1 - \alpha_2)\left(2.15 - 3.93\alpha_2 + 2.7\alpha_2^2\right)}{\sqrt{\pi}(1 + 2\alpha_2)(1 - \alpha_2)^{3/2}}, \tag{2}$$

$$\alpha_2 = \frac{a_c}{d}, \ w = \frac{W_0 S}{L}, \text{ N} \tag{3}$$

where: $P_{max}$ - maximum load, $W_0$ - specimen weight, N and $S$, $L$, $a_o$, $d$, $b$ (according to Figure 2).

The mean values of $K_{Ic}{}^S$ were calculated on the basis of four results.

The fractal tests were performed on specially prepared gypsum replicas of fracture surfaces (Figure 2). The replicas made from white and dyed gypsum were cut along the longer edge of the block into layers (Figure 2b), which were next scanned at the resolution of 600 dpi. By means of FRACTAL_Digit (J. Konkol, FRACTAL_Digit, a program, 2001), an image of the profile line was obtained, which is the line separating the white and contrasting gypsum layers, as shown in Figure 2c. To calculate the fractal dimension of the profile line $D_{BC}$, the box counting method was applied. The fractal analysis of concrete fracture surface profile lines was done with FRACTAL_Dimension2D (J. Konkol, FRACTAL_Dimension2D, a program, 2000). The fractal dimension was determined as the tangent of the slope in a double-logarithmic diagram of the relationship of the logarithm of the box number to the logarithm of the box dimension.

The stereological tests were done on polished plane cross-sections of concrete test specimens. In total, 450 plane cross-sections were prepared. After grinding the concrete section, the areas were blackened and the pores were filled with zinc paste.

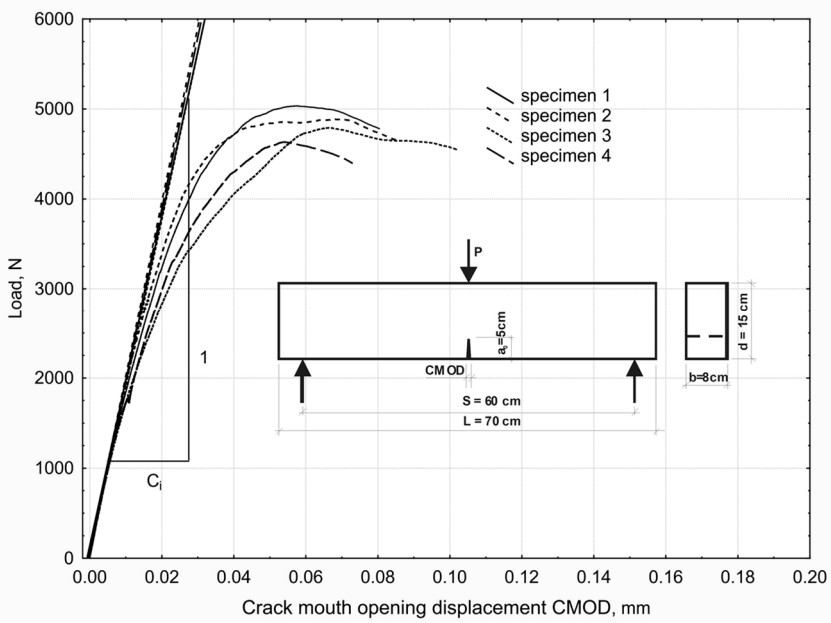

**Figure 1.** An example graph of crack mouth opening displacement (CMOD)–load relationship and schematic drawings of the specimen used in the fracture toughness examination according to Mode I (concrete with fluidal ash additive–series 5).

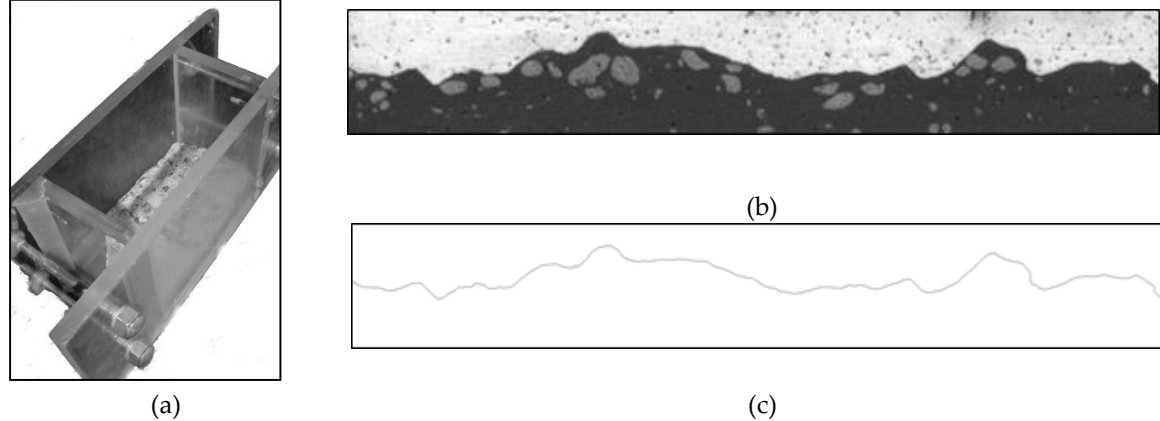

**Figure 2.** A mold for the manufacture of gypsum replicas (**a**) and a gypsum replica strip (**b**) scanned at definition resolution of 600 dpi together with the digitalization results (**c**).

The images of air pores were scanned at the resolution of 400 dpi. Computer image analysis of the bitmaps obtained was performed next. Real images of the test specimen surfaces and the results of the transformations are shown in Figure 3. The stereological analysis of the air pores was carried out by means of FRACTAL,_Stereology (J. Konkol, FRACTAL_Stereology, a program, 2002). Figure 4 shows how the program interpreted the locations of occurrence of air pores in the greyscale range of 170–255, which was determined in the preliminary analyses.

On the basis of the stereological measurements, the air pore relative area $S_{VP}$ was calculated. This surface, which is a measure of dispersion, was calculated as the total surface of pores by volume unit. The stereological analysis was also done for the stage of coarse aggregate, determining the relative area of coarse aggregate $S_{VK}$.

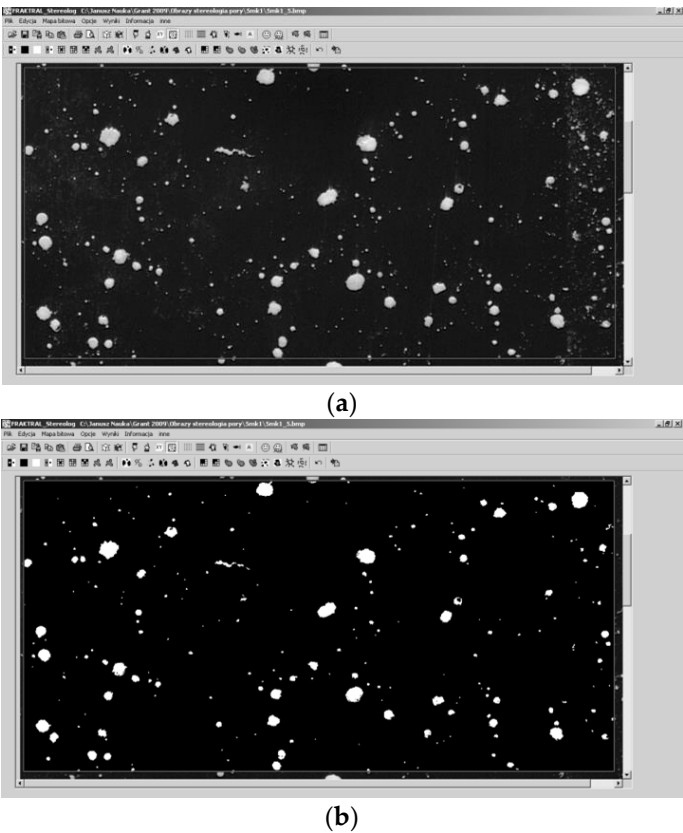

(**a**)

(**b**)

**Figure 3.** Test specimens surface with visible pores filled with zinc paste (**a**), after treatment filling—white spots represent places identified as air pores sections (**b**).

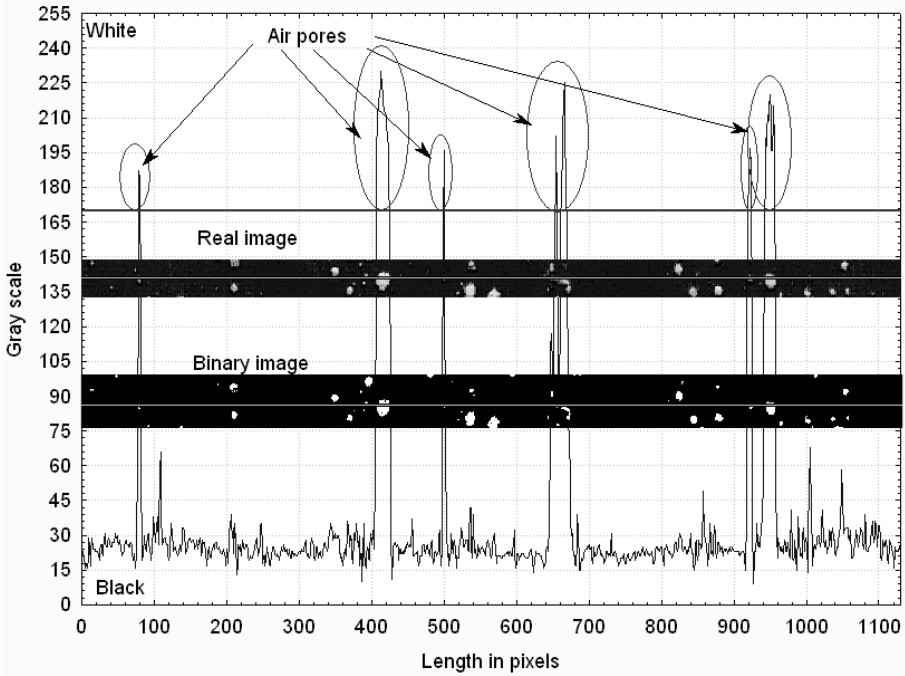

**Figure 4.** Method of identification of places of air pores occurrence (greyscale for pores from 170 to 255).

## 3. Results

The results of tests on compressive strength and fracture toughness are given in Table 2. Table 3 presents the results of the fractal analyses of the profile lines and stereological of the pores.

**Table 2.** Results of tests on compressive strength $f_c$ and critical stress intensity factor $K_{Ic}{}^S$ after 28 days curing of concrete.

| Series No. | Mechanical Properties of Modified Concrete | | | | | |
| --- | --- | --- | --- | --- | --- | --- |
| | Activated Fluidal Ash (FA) | | Metakaolinite (MK) | | Silica Fume (SF) | |
| | $f_c \pm$ Stand. Error MPa | $K_{Ic}{}^S \pm$ Stand. Error MN/m$^{3/2}$ | $f_c \pm$ Stand. Error MPa | $K_{Ic}{}^S \pm$ Stand. Error MN/m$^{3/2}$ | $f_c \pm$ Stand. Error MPa | $K_{Ic}{}^S \pm$ Stand. Error MN/m$^{3/2}$ |
| 1 | 58.3 ± 1.2 | 1.49 ± 0.03 | 53.7 ± 0.5 | 1.44 ± 0.04 | 54.8 ± 1.1 | 1.25 ± 0.04 |
| 2 | 61.0 ± 1.1 | 1.58 ± 0.03 | 61.0 ± 1.2 | 1.57 ± 0.02 | 65.3 ± 1.1 | 1.53 ± 0.05 |
| 3 | 40.0 ± 0.9 | 0.90 ± 0.05 | 40.8 ± 1.2 | 0.94 ± 0.03 | 39.2 ± 0.7 | 0.92 ± 0.02 |
| 4 | 40.9 ± 0.6 | 1.17 ± 0.03 | 41.3 ± 0.4 | 1.02 ± 0.06 | 40.2 ± 1.3 | 0.97 ± 0.03 |
| 5 | 63.8 ± 0.4 | 1.47 ± 0.05 | 63.7 ± 0.6 | 1.52 ± 0.03 | 66.1 ± 0.7 | 1.58 ± 0.01 |
| 6 | 41.5 ± 1.4 | 1.06 ± 0.05 | 37.2 ± 0.8 | 0.97 ± 0.01 | 38.6 ± 0.5 | 0.93 ± 0.03 |
| 7 | 45.2 ± 0.8 | 1.23 ± 0.04 | 46.7 ± 0.5 | 1.25 ± 0.05 | 46.7 ± 1.0 | 1.24 ± 0.06 |
| 8 | 47.3 ± 0.9 | 1.34 ± 0.10 | 51.5 ± 0.9 | 1.32 ± 0.02 | 54.8 ± 1.1 | 1.34 ± 0.04 |
| 9 | 45.5 ± 1.0 | 1.27 ± 0.04 | 47.8 ± 1.0 | 1.25 ± 0.06 | 48.8 ± 0.7 | 1.21 ± 0.03 |
| 10 | 45.9 ± 1.1 | 1.25 ± 0.12 | 48.0 ± 1.0 | 1.20 ± 0.04 | 49.2 ± 0.5 | 1.22 ± 0.01 |

**Table 3.** Results of the fractal and stereological testing.

| Series No. | Fractal and Stereological Parameters of Modified Concrete | | | | | |
| --- | --- | --- | --- | --- | --- | --- |
| | Activated Fluidal Ash (FA) | | Metakaolinite (MK) | | Silica Fume (SF) | |
| | $D_{BC} \pm$ Stand. Error - | $S_{VP} \pm$ Stand. Error cm$^2$/cm$^3$ | $D_{BC} \pm$ Stand. Error - | $S_{VP} \pm$ Stand. Error cm$^2$/cm$^3$ | $D_{BC} \pm$ Stand. Error - | $S_{VP} \pm$ Stand. Error cm$^2$/cm$^3$ |
| 1 | 1.047 ± 0.001 | 2.69 ± 0.09 | 1.051 ± 0.001 | 2.36 ± 0.08 | 1.046 ± 0.001 | 2.75 ± 0.08 |
| 2 | 1.044 ± 0.001 | 2.38 ± 0.11 | 1.045 ± 0.001 | 2.32 ± 0.08 | 1.047 ± 0.001 | 2.56 ± 0.08 |
| 3 | 1.054 ± 0.001 | 2.35 ± 0.20 | 1.053 ± 0.001 | 1.68 ± 0.08 | 1.051 ± 0.001 | 2.73 ± 0.10 |
| 4 | 1.050 ± 0.001 | 2.49 ± 0.17 | 1.047 ± 0.001 | 1.97 ± 0.09 | 1.054 ± 0.001 | 2.64 ± 0.07 |
| 5 | 1.047 ± 0.001 | 2.62 ± 0.12 | 1.045 ± 0.001 | 2.17 ± 0.12 | 1.042 ± 0.001 | 2.49 ± 0.09 |
| 6 | 1.051 ± 0.001 | 2.09 ± 0.08 | 1.050 ± 0.001 | 1.91 ± 0.10 | 1.052 ± 0.001 | 1.65 ± 0.05 |
| 7 | 1.050 ± 0.001 | 2.27 ± 0.09 | 1.051 ± 0.001 | 2.48 ± 0.11 | 1.049 ± 0.001 | 2.46 ± 0.09 |
| 8 | 1.047 ± 0.001 | 2.75 ± 0.10 | 1.047 ± 0.001 | 1.80 ± 0.07 | 1.050 ± 0.001 | 2.46 ± 0.08 |
| 9 | 1.050 ± 0.001 | 2.59 ± 0.14 | 1.049 ± 0.001 | 2.13 ± 0.09 | 1.048 ± 0.001 | 2.42 ± 0.08 |
| 10 | 1.047 ± 0.002 | 2.16 ± 0.12 | 1.050 ± 0.001 | 1.95 ± 0.16 | 1.048 ± 0.001 | 2.46 ± 0.17 |

Mean values of fractal dimension $D_{BC}$ calculated on the basis of the analysis of 16–22 profile lines, mean values of pore relative area $S_{VP}$ based on 12 images of 25 cm$^2$ each.

Tests on compressive strength and fracture toughness have also been performed on some concretes with no additives. The comparison of results for concretes with no additives and those modified is shown in Figures 5 and 6.

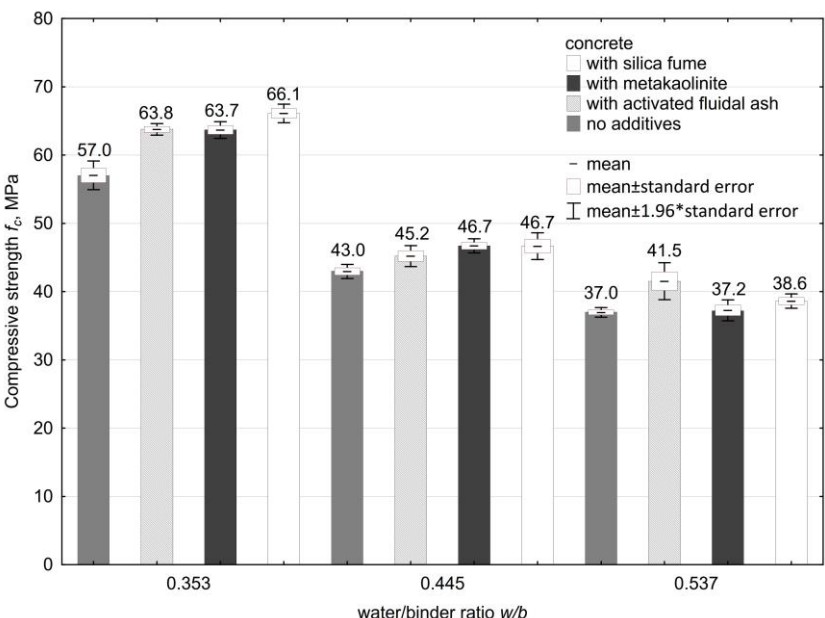

**Figure 5.** Compressive strength of concretes with no additives versus modified concretes (at $w/b$ = 0.353; 0.445, and 0.537 additive content as for concrete series 5, 7 and 6 adopted in the experiment plan).

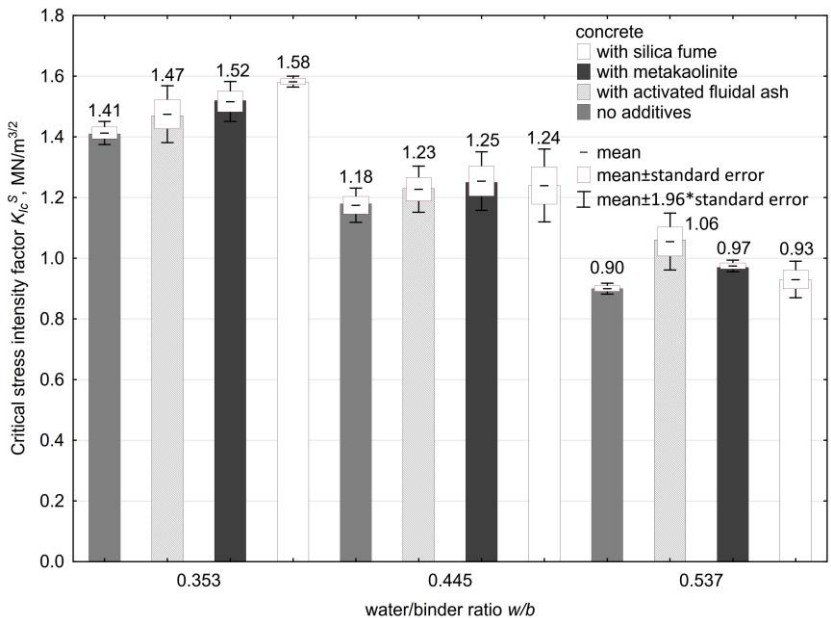

**Figure 6.** Critical stress intensity factor $K_{Ic}^{S}$ of concretes with no additives versus modified concretes (at $w/b$ = 0.353; 0.445, and 0.537 additive content as for concrete series 5, 7 and 6 adopted in the experiment plan).

## 4. Discussion

The application of additives resulted in improved compressive strength and a higher critical stress intensity factor. The highest effectiveness of all the additives was observed for concretes of the lowest water/binder ratio ($w/b$). The higher the water/binder ratio, the lower the effect of the additives on the compressive strength and fracture toughness.

By analyzing the compressive strength and critical stress intensity factor for a given type of concrete (concrete without additive or concrete modified with the given additive, Figures 5 and 6), a statistically significant effect of the change in the water/binder ratio on these properties was found

(Table 4, obtained values of the limit significance level $p < 0.05$). The analysis was carried out using the Snedecor–Fisher F-test, which is an equality test for average values.

**Table 4.** Analysis of variance.

| Concrete Type | Equality Test for Average Values | | | |
| | in the Case of Compressive Strength $f_c$ | | in the Case of the Critical Stress Intensity Factor $K_{Ic}{}^S$ | |
| | The Value of the F Test | Limit Level of Significance of the Test | The Value of the F Test | Limit Level of Significance of the Test |
|---|---|---|---|---|
| no additives | 205.6 | close to 0 | 152.5 | close to 0 |
| with MK | 436.3 | close to 0 | 60.2 | close to 0 |
| with FA | 176.7 | close to 0 | 21.9 | 0.0003 |
| with SF | 388.6 | close to 0 | 66.7 | close to 0 |
| **Water/binder ratio $w/b$** | **The value of the F test** | **Limit level of significance of the test** | **The value of the F test** | **Limit level of significance of the test** |
| | all concretes | | | |
| 0.353 | 27.1 | close to 0 | 5.21 | 0.0156 |
| 0.445 | 6.64 | 0.0018 | 0.56 | 0.6486 |
| 0.537 | 6.57 | 0.0019 | 5.35 | 0.0142 |
| | only concretes with additives | | | |
| 0.353 | 5.41 | 0.0127 | 2.51 | 0.1364 |
| 0.445 | 1.28 | 0.3023 | 0.07 | 0.9328 |
| 0.537 | 5.11 | 0.0175 | 3.63 | 0.0698 |

Except for one case (for $K_{Ic}{}^S$ and $w/b = 0.445$, Table 4), the use of the F-test for analysis of test results obtained within concrete with a constant water/binder ratio indicated the difference between the mean values of compressive strength and critical stress intensity factor of these concretes.

The elimination of the results of the critical stress intensity factor obtained for concrete without additive and re-analysis of the mean equality indicated the lack of statistically significant differences between the mean values of the critical stress intensity factor $K_{Ic}{}^S$ of the concretes modified with the selected additive (Table 4).

In the case of the compressive strength test results, only for the case $w/b = 0.445$ (Table 4), there was no reason to reject the hypothesis of equality of the means.

Preceding further statistical analysis for all the results of the tested parameters covered by the research plan, the homogeneity of variance and significance of the impact of the adopted variables on the tested parameters were checked. In each case, the homogeneity of variance was demonstrated together with a statistically significant (at the level of significance 0.05) influence of variables on the studied parameters. Since the relation between the compressive strength and fracture toughness was known, a correlation analysis was done, at a significance level of 0.05. The analysis was done for all of the modified concretes, disregarding the type of additive, since the analysis of multiple regression proved the insignificance of the variable of the additive type.

The application of all three types of dusty mineral additives has improved both compressive strength and fracture toughness. The increase of both characteristics can be described by a linear relation, Equation (4):

$$K_{Ic}^S = 0.025 \cdot f_c \tag{4}$$

The model correlation coefficient, Equation (4), was 0.931, and the coefficient of determination was 0.867. The variations of the critical stress intensity factor $K_{Ic}{}^S$ was then accounted for by the variation of the compressive strength $f_c$ in 86.7% and the variation of other factors, including the random ones in 13.3%.

Figure 7 shows a plot of the values observed versus those predicted $K_{Ic}{}^S$. The points corresponding to the results for concretes modified with Flubet (FA), metakaolinite (MK) and silica fume (SF) have been marked.

Referring to the relation between elastic modulus $E$ and concrete compressive strength $f_c$, known from the literature, a statistical analysis was done for the model:

$$E = a \cdot f_c^b \tag{5}$$

The values $a = 4.709$ and $b = 0.509$ were obtained (Figure 8). The significance of the coefficients of Equation (5) was proven and the correlation coefficient $R$ was 0.810. The solution reached was close to a known relation for normal concrete, recommended by ACI 318-89.

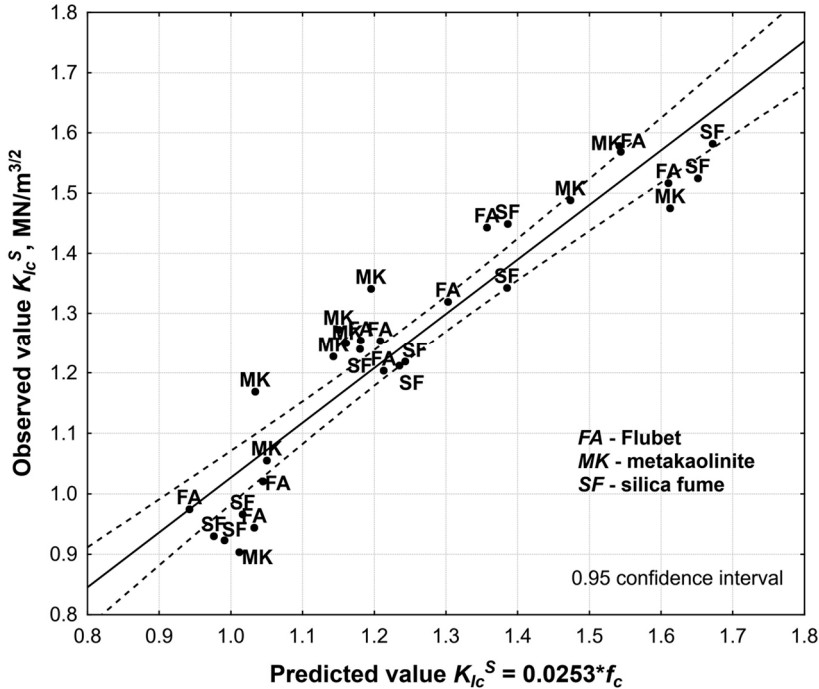

**Figure 7.** Relation between the values of critical stress intensity factor $K_{Ic}^S$ observed and that determined with Equation (4).

Further analyses showed the relationship between the critical stress intensity factor $K_{Ic}^S$ and the water/binder ratio $w/s$ affecting the class of concrete strength, and the fractal dimension $D_{BC}$, which characterizes quantitatively the fracture surface during the cracking process. The analysis was done by means of a multiple regression method (Table 5), showing the significance of the effect of both values ($w/b$ and $D_{BC}$) on the change of factor $K_{Ic}^S$. An improvement of the correlation coefficient $R = 0.931$ for Equation (1) to 0.960 for Equation (6) was obtained:

$$K_{Ic}^S = 20.164 - 2.869 \cdot w/b - 16.813 \cdot D_{BC} \tag{6}$$

The absolute term in Equation (6) and other terms of the multiple regression equation were statistically significant on the adopted significance level of 0.05 ($p < 0.05$, Table 5). The variation of the critical stress intensity factor $K_{Ic}^S$ was thus accounted for by the variation of the water/binder ratio and the fractal dimension in 92.2%, and only 7.8% were other factors, including the random ones. At the same time, the values of the standardized coefficients of regression $b^*$ (Table 5) indicated the $w/b$ variable contribution was higher by 3.5 than that of variable $D_{BC}$ in the prediction $K_{Ic}^S$. Figure 9 illustrates a plot of the values observed versus the predicted $K_{Ic}^S$, calculated on the basis of Equation (6).

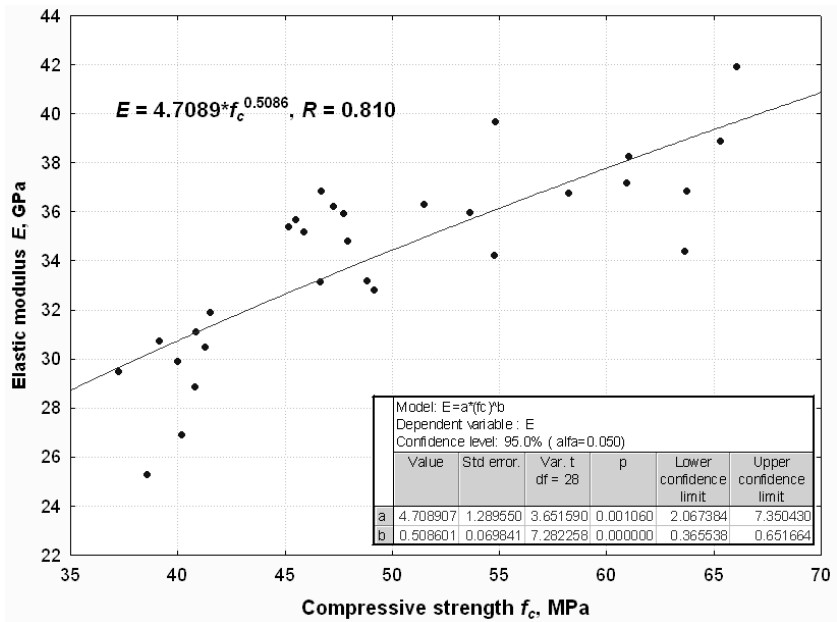

**Figure 8.** Exponential dependence between elastic modulus $E$ and compressive strength $f_c$ of modified concretes.

**Table 5.** Summary of multiple regression analysis in case of Equation (6).

| N = 30 | Summary Regression of the Dependent Variable $K_{Ic}{}^S$ $R = 0.960$, $R^2 = 0.922$ $F (2.27) = 159.68$, $p < 0.00000$, Std. Error Estimation: 0.0625 | | | | | |
|---|---|---|---|---|---|---|
| | $b^*$ | **Stand. Error** $b^*$ | $b$ | **Stand. Error** $b$ | $t(27)$ | $p$ |
| Absolute term | | | 20.161 | 6.3245 | 3.188 | 0.0036 |
| $w/b$ | −0.7854 | 0.0802 | −2.869 | 0.2929 | −9.797 | 0.0000 |
| $D_{BC}$ | −0.2202 | 0.0802 | −16.813 | 6.1224 | −2.746 | 0.0106 |

To predict the values of $K_{Ic}{}^S$ on the basis of Equation (6), it is necessary to know the fractal dimension, which is connected with the necessity of performing destructive testing and analyzing the fracture surface formed.

As is well known, however, a fracture in concrete results from tensile stress exceeding its strength. The fracture surface that is formed and its separated part (the profile line), can have a different shape and roughness. The factors that affect the profile line shape roughness are the element phases of the concrete composition, such as the phase of aggregate, the phase of hardened cement paste or mortar, and the phases of the defects (e.g., pores).

To determine which of the variables affect, to the highest extent, the prediction of the fractal dimension $D_{BC}$, the technique of linear multiple regression was applied. Stereological parameters of the aggregate relative area $S_{VK}$ and the pore relative area $S_{VP}$ were adopted as the variables describing the phases of aggregate and pores (Table 3). Additionally, the additive to binder ratio $AD/b$, volume of cement paste $V_{Paste}$ and type of additive, as a qualitative variable, were adopted as variables.

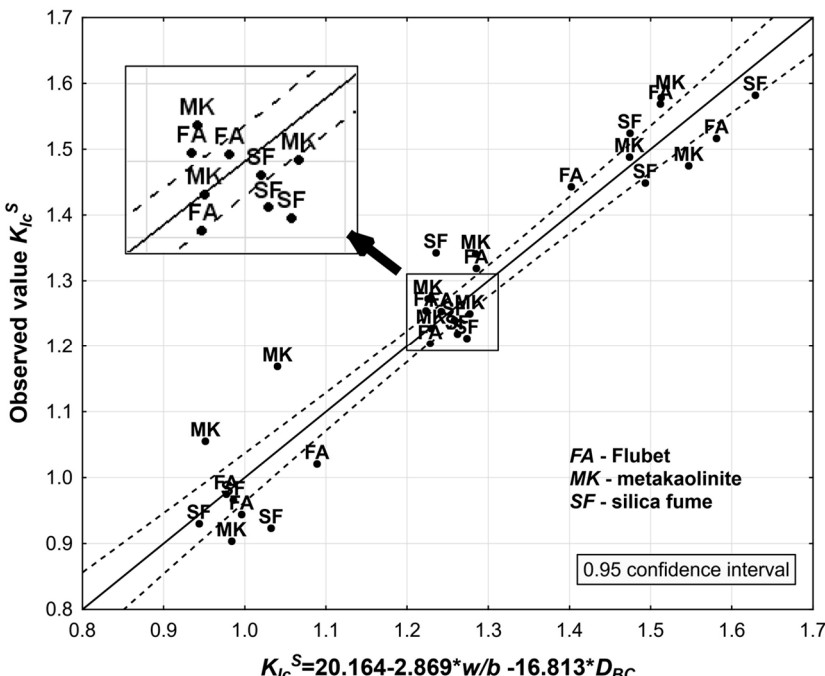

**Figure 9.** Relation between critical stress intensity factor $K_{Ic}^S$ observed and that determined with Equation (6).

On the basis of statistical analysis, the variables were considered insignificant (on a significance level of 0.05), such as the aggregate relative area $S_{VK}$, pore relative area $S_{VP}$ and type of additive ($p > 0.05$, Table 6) were rejected. The model of multiple regression obtained the final form:

$$D_{BC} = 1.088 - 0.026 \cdot AD/b + 0.123 \cdot V_{Paste} \tag{7}$$

where: $AD/b$—additive mass (FA, MK or SF) to binder mass ratio; $V_{Paste}$—content of cement paste in concrete mix.

**Table 6.** Results of multiple regression after removal of insignificant effects.

| N = 30 | Summary Regression of the Dependent Variable $D_{BC}$<br>$R = 0.811$, $R^2 = 0.657$<br>$F (2.27) = 25.870$, $p < 0.00000$, Std. Error Estimation: 0.0017 | | | | | |
| --- | --- | --- | --- | --- | --- | --- |
|  | $b^*$ | Stand. Error $b^*$ | $b$ | Stand. Error $b$ | $t(27)$ | $p$ |
| Absolute term |  |  | 1.088 | 0.0063 | 159.25 | 0.0000 |
| $AD/b$ | −0.3567 | 0.1129 | −0.026 | 0.0082 | −3.161 | 0.0039 |
| $V_{Paste}$ | 0.7470 | 0.1129 | 0.123 | 0.0185 | 6.620 | 0.0000 |

An important conclusion to make is the fractal dimension, independent of additive type.

The equation of Equation (7) was highly significant ($R = 0.811$ i $p \approx 0.000$). The values of standardized coefficients of regression $b^*$ (Table 6) indicated that the concrete paste $V_{Paste}$ contributed to the prediction of the fractal dimension twice as high as the additives ($AD/b$). The plot of the observed values $D_{BC}$ versus the predicted ones (Equation (7)) is shown in Figure 10.

As was proved, in the case of the fractal examinations, the fractal dimension results scattering was significant, which led to the lowering of the correlation coefficient calculated on the basis of single measurements. The credibility of the obtained results was confirmed by both statistical analyses and the assumptions for performing fractal examinations using an appropriate number of profile lines (at

least 12, according to Konkol and Prokopski [19]). Such an approach led to obtaining a result of an average value with an assumed estimation error.

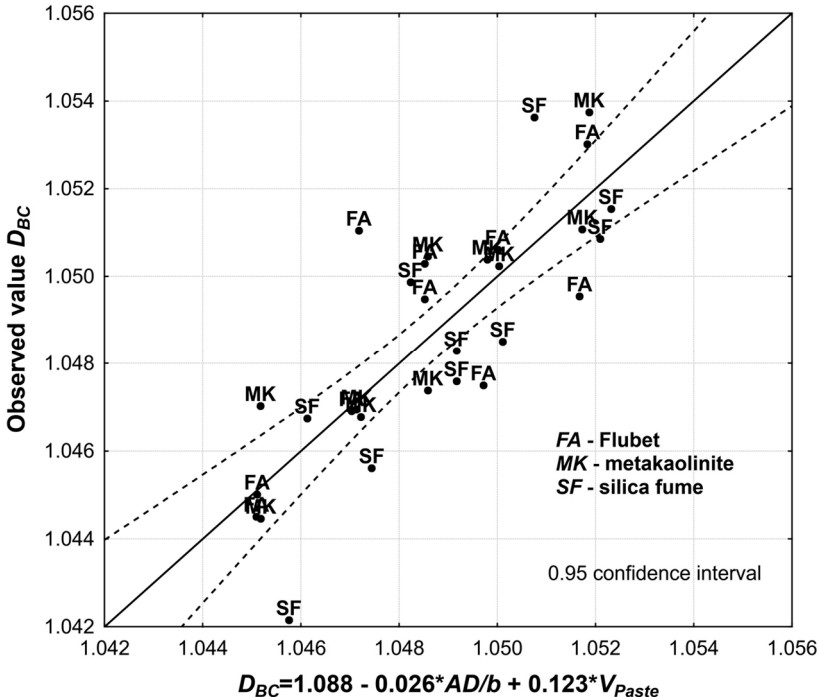

**Figure 10.** Relation between fractal dimension $D_{BC}$ observed and that determined with Equation (7).

In other research done by the author on concretes with no additives, the insignificance of the pore relative area $S_{VP}$ has also been shown. This insignificance also was confirmed in research on concretes modified with activated fluidal ash or metakaolinite [55]. The insignificance of the aggregate relative area $S_{VK}$ can be accounted for by small variation of this parameter, since the content of this aggregate was not a variable included in the experiment plan adopted. The proportion of the coarse aggregate was from 37 to 41% of the concrete volume.

An increased proportion of additive results in a higher density of concrete paste composition with simultaneous smoothing of the profile line through the paste, which reduces the fractal dimension $D_{BC}$. An increased proportion of the cement paste in concrete, on the other hand, enlarges the fractal dimension $D_{BC}$. This phenomenon can be explained by a higher roughness of the hardened cement paste on the fracture, compared with the roughness of the basalt grains. This is because a higher content of the paste results in a lower content of basalt aggregate.

When the two proposed models (Equations (6) and (7)) are combined, it is possible, on the basis of concrete composition, to determine the fracture toughness of concretes modified with the additives applied without the necessity of destructive testing.

Based on the analysis of the test results, it was found that the error of the suggested solution, based on Equations (6) and (7), was lower than the differences resulting from the scatter of the individual results around an average value. The mean error of $K_{Ic}^{S}$ estimated was 4.0 %, the extreme values reached –12.1% and +10.4%. In 87% of the results, the $K_{Ic}^{S}$ estimate error was in the range of –8% to +8%. The distribution of the $K_{Ic}^{S}$ estimate error is shown in Figure 11. On the other hand, the mean difference between an individual $K_{Ic}^{S}$ result and an average mean at the given point of the experiment plan was 4.9%, with the extreme values of +19.7% and –14.9% (Figure 12).

A graphic comparison of the $K_{Ic}^{S}$, results, calculated on the basis of Equations (6) and (7), with the values of $K_{Ic}^{S}$ obtained in testing is in Figure 13. The analysis of linear regression indicated that the observed values of $K_{Ic}^{S}$ versus the predicted ones were distributed along a straight line, with a

significant slope of 1.0. The result implied good compatibility of the predicted values with the observed ones. It also confirmed the reliability of the proposed solution.

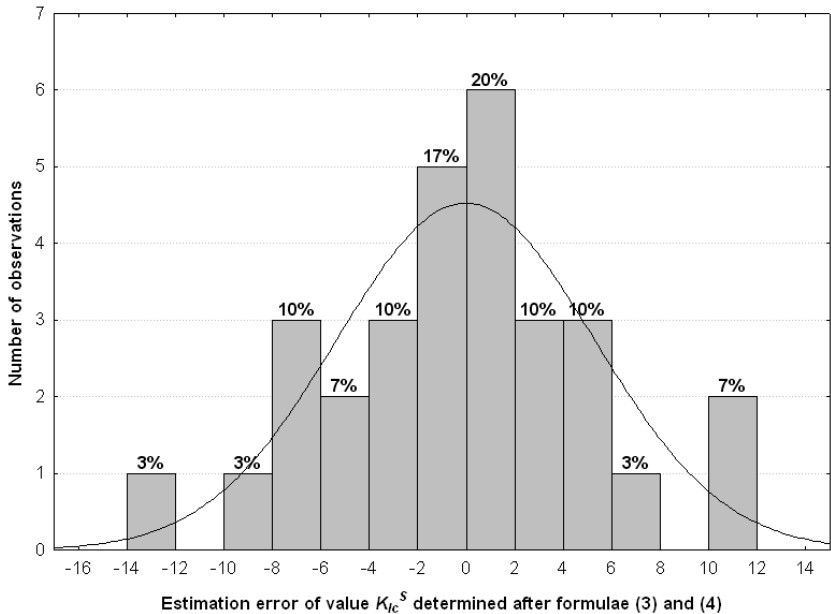

**Figure 11.** Histogram of $K_{Ic}^{S}$ estimation error distribution determined after Equations (6) and (7) compared with $K_{Ic}^{S}$ values determined in testing.

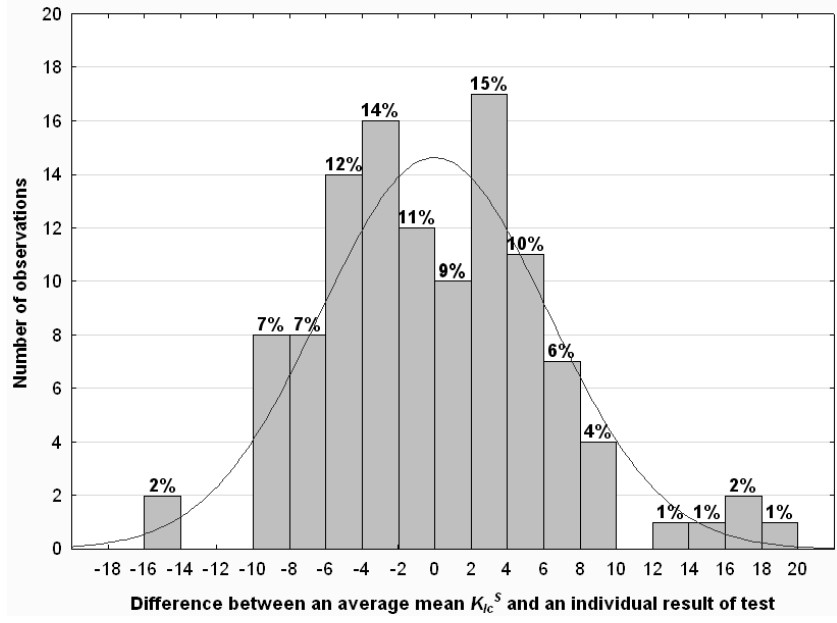

**Figure 12.** Histogram of distribution of differences between mean values of $K_{Ic}^{S}$, determined for each series of concretes, and values of $K_{Ic}^{S}$ determined on the basis of individual results of tests.

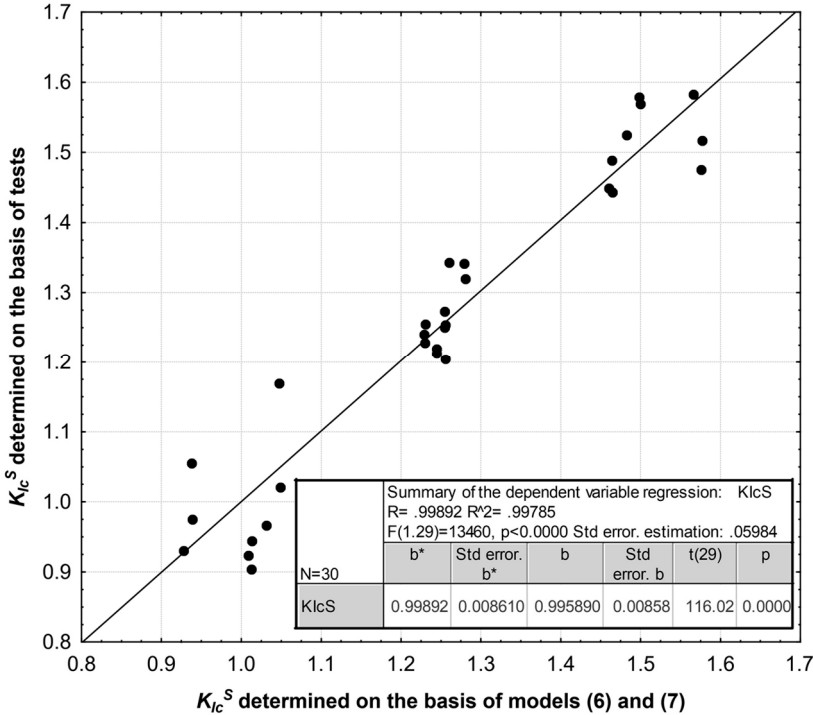

**Figure 13.** $K_{Ic}^{S}$ observed values versus predicted ones, determined on the basis of Equations (6) and (7).

## 5. Conclusions

The solution presented in the article, based on Equations (4)–(7), enabled the prediction of the properties of dusty additive-modified concretes (silica fume, activated fluidal ash or metakaolinite) after 28 days of curing, without the necessity of laboratory testing.

To determine the compressive strength $f_c$, fracture toughness (determination of critical stress intensity factor $K_{Ic}^{S}$), and elastic modulus in bending $E$, it was sufficient to define the additive to binder mass ratio $AD/b$ (the binder side additive is also included), water/binder ratio $w/b$ ($w/b = w/(c + AD)$) and the content of concrete paste in concrete mix $V_{Paste}$.

In Equations (6) and (7), the analysis included the parameters characterizing the concrete composition (the stereological parameters) and the fracture surface morphology (the fractal dimension).

The statistical analyses led to an important conclusion, that there was no significant effect of the type of additive on the parameters in question. In Equations (6) and (7), which enabled the critical stress intensity factor $K_{Ic}^{S}$ to be determined, it was proved that the calculation error for $K_{Ic}^{S}$ was smaller than the difference resulting from the scatter of the individual results of tests on $K_{Ic}^{S}$. Moreover, the linear relationship, which was statistically highly significant at the adopted significance level of 0.05, between the critical stress intensity factor $K_{Ic}^{S}$ and compressive strength $f_c$ of modified concretes after 28 days of curing was shown. Equation (6) enabled the elasticity modulus in bending $E$ to be determined.

Equations (4)–(7) were valid for concretes made from Portland cement CEM I 32.5R and coarse basalt aggregate with a water/binder ratio $w/b$ in the range of 0.35 to 0.54. The maximum content of additives could amount up to 17.5% of cement mass for mechanically activated fluidal ash or metakaolinite, and up to 11.4% of cement mass for silica fume.

The proposed solution, with fracture toughness taken into account, is an important contribution in the design of concretes modified with three additives separately: Silica fume, activated fluidal ash or metakaolonite, and can be a useful tool for a concrete design engineer.

The statistically significant dependence between fracture toughness and modified concrete structure, including the fracture surface morphology, described by the fractal dimension (Equation (6)), confirms the need of research targeted at improved knowledge and modeling of concrete fracture processes with fractal geometry taken into consideration.

**Author Contributions:** Conceptualization, J.K.; Methodology, J.K.; Software, J.K.; Validation, J.K.; Formal Analysis, J.K.; Investigation, J.K.; Resources, J.K.; Data Curation, J.K.; Writing-Original Draft Preparation, J.K.; Writing-Review & Editing, J.K.; Visualization, J.K.; Supervision, Grzegorz Prokopski; Project Administration, J.K.; Funding Acquisition, J.K.

**Funding:** This research was funded by the Polish Grant Agency MNiSzW, grant number N N507 475337

**Acknowledgments:** The author would like to appreciate the anonymous reviewers for their constructive suggestions to improve the quality of the paper.

**Conflicts of Interest:** The author declares no conflict of interest.

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
