# Peer review of "Fracture Toughness and Fracture Surface Morphology of Concretes Modified with Selected Additives of Pozzolanic Properties"

_buildings, doi:10.3390/buildings9080174_

Reviewer 1 Report

The reviewer has the following comments:

Provide the detailed properties of additives, i.e. silica fume, ash, and metakaolin.

Usually, figures are not included in Introduction. (Fig. 1). Also provide the details of aggregate used.

How was the range of w/b ratio and replacement ratios of the additives determined?

Fracture properties are largely affect by the size (dimensions) and specimen shape. How was the size effected treated for model generalization?

Is Fig. 2 necessary?

The models developed can be only applied to the mixture the authors tested. The model cannot be generalized.

Provide error bars in Fig. 6.

How many specimens were tested to measure critical stress intensity factors?

What is the novelty of this paper? Already many research papers published in this matter over the years.

Author Response

Response to Reviewer 1 Comments

Thank you for the insightful review of the submitted manuscript and the suggestions of the Reviewers for its improvement.

In the revised manuscript, all comments from the Reviewers were taken into account.

 Answers to comments

Reviewer 1

1. Provide the detailed properties of additives, i.e. silica fume, ash, and metakaolin.

Lines 74-87 –  In response to the Reviewer’s comment the text of the article has been supplemented with details of the additives used.

2. Usually, figures are not included in Introduction. (Fig. 1). Also provide the details of aggregate used.

As suggested by the Reviewer, Figure 1 has been removed.

Lines 72-73 – Information on the aggregates used has been added.

3. How was the range of w/b ratio and replacement ratios of the additives determined?

The range of w/b ratio and replacement ratio of the additives was determined experimentally.

While taking the share of the additive, present knowledge as well as the manufacturer’s data and personal experience were used as guidelines. Due to the purpose of the research, among others, to disseminate the use of the metakaolinite additive, the w / b range was adopted as for concretes made on an industrial scale.

The use of the experiment plan enforced limitations in the selection of the scope of both variables, due to the inability to control the consistency and workability of concrete mixes, for mixtures outside the adopted plan. The accepted range of the share of additives also took into account the possibility of their pozzolanic reaction with Ca (OH) 2. Too large a share could cause the part of the additive to be unresponsive and only fulfill the filler function.

 4. Fracture properties are largely affect by the size (dimensions) and specimen shape. How was the size effected treated for model generalization?

The results of the fracture toughness test as well as the compressive strength depend on the size and shape of the samples. The research presented in the article was conducted in accordance with the draft RILEM recommendations and concerns the first cracking model.

This procedure is widely used and recommended for use in mortars and concretes. Depending on the grain size of the coarse aggregate, samples with given dimensions are recommended - for aggregates over 16 mm, the size and shape of the samples correspond to the samples used in the tests. The proposed model is based on the assumption of complying with the RILEM recommendations, which enables generalization of the model.

5. Is Fig. 2 necessary?

Figure 2 in addition to the presentation of an example graph of CMOD-load relationship also shows schematic drawings of the specimen used in the fracture toughness examination according to Mode I. This is important because the studies could be conducted with other types of samples.

The author proposes to leave Figure 2.

6. The models developed can be only applied to the mixture the authors tested. The model cannot be generalized.

Of course, the use of other components of the concrete mix is an obstacle to the application of the proposed model. However, the acceptance of the components used in the research (commonly used) allows the use of the model presented in the article. A generalization is the possibility of describing a given concrete property with accepted variables. At the same time, tests of concretes made on the same components with the replacement of only the additive, as comparative tests, supplement the knowledge in this area, the more so that the tests of resistance to cracking in such a range have not been carried out so far.

7. Provide error bars in Fig. 6.

The Reviewer’s comment was taken into account. Figures 6 and 7 have been supplemented with error bars.

 8. How many specimens were tested to measure critical stress intensity factors?

Information on the number of repetitions within each series of concretes (number of specimens) is given in the text of the manuscript.

Currently line 112. “Mean values of KIcS calculated on the basis of 4 results.”

9. What is the novelty of this paper? Already many research papers published in this matter over the years.

The fracture toughness tests for concrete are still uncommon. The need to conduct such research is indisputable. This applies first of all to materials with a cement matrix modified with pozzolanic additives. Lowering fracture toughness can often cause serious problems for concrete structures. Tests of fracture toughness of metakaolinite modified concretes have not been carried out in such a wide range so far, let alone comparing the obtained results with modified concrete with the already known silica fume or a new kind of ash (fluidal ash).

At the same time, the obtained results show the possibilities of combining, in accordance with the material engineering approach, the properties of these concretes with their structure. They are proof that once abandoned attempts at a fractal approach, can be resumed. It turns out that linking the fractal dimension to a given property may be impossible, as indicated by previous publications. As research and analysis have shown, this approach requires the introduction of other parameters describing the structure of the concrete. A novelty in the complex manuscript is the use of both the statistical apparatus and the theory of experiment planning to achieve the assumed goal. Interest in this subject is confirmed by the author's publications, among others Konkol, J.; Prokopski, G. Fracture toughness and fracture surfaces morphology of metakaolinite-modified concrete. Constr Build Mater, 2016, 123, 638–648, doi.org/10.1016/j.conbuildmat.2016.07.025.

Reviewer 2 Report

This manuscript presents comparative study on Fracture toughness of concretes modified with pozzolanic additives.

The topic is interesting and the presented results are promising. 

The following comments should be considered to improve 

this paper before its acceptance in this journal:

Statistical analysis should be extended to include better comparison between the results, e.g. using t-test to indicate if the differences are statistical significant. For example, Figs 6 and 7 have no error bars, so the difference is difficult to judge, in terms of measurement uncertainty. Please provide error bars and discuss the results in terms of  statistical significant differences.

Author Response

Response to Reviewer 2 Comments

Thank you for the insightful review of the submitted manuscript and the suggestions of the Reviewers for its improvement.

In the revised manuscript, all comments from the Reviewers were taken into account.

 Answers to comments

Reviewer 2

This manuscript presents comparative study on Fracture toughness of concretes modified with pozzolanic additives. The topic is interesting and the presented results are promising. The following comments should be considered to improve this paper before its acceptance in this journal:

Statistical analysis should be extended to include better comparison between the results, e.g. using t-test to indicate if the differences are statistical significant. For example, Figs 6 and 7 have no error bars, so the difference is difficult to judge, in terms of measurement uncertainty. Please provide error bars and discuss the results in terms of statistical significant differences.

The Reviewer’s comment was taken into account. Figures 6 and 7 have been supplemented with error bars. An appropriate comment was added to the drawings in the manuscript text - the results in terms of statistical significant differences were discussed.

Analyzing the compressive strength and critical stress intensity factor for a given type of concrete (concrete without additive or concrete modified with the given additive, Figures 5 and 6), a statistically significant effect of the change in the water/binder ratio on these properties was found (Table 4, obtained values of the limit significance level <<0.05). The analysis was carried out using the Snedecor-Fisher F test - an equality test for average values.

Table 4. Analysis of variance

Concrete type

Equality test for average values

in the case   of compressive strength fc

in the case   of the critical stress intensity factor KIcS

The value of   the F test

Limit level   of significance of the test

The value of   the F test

Limit level   of significance of the test

no additives

205.6

close to 0

152.5

close to 0

with MK

436.3

close to 0

60.2

close to 0

with FA

176.7

close to 0

21.9

0.0003

with SF

388.6

close to 0

66.7

close to 0

Water/binder   ratio w/b

The value of the F test

Limit level of significance of the test

The value of the F test

Limit level of significance of the test

all   concretes

0.353

27.1

close to 0

5.21

0.0156

0.445

6.64

0.0018

0.56

0.6486

0.535

6.57

0.0019

5.35

0.0142

only   concretes with additives

0.353

5.41

0.0127

2.51

0.1364

0.445

1.28

0.3023

0.07

0.9328

0.535

5.11

0.0175

3.63

0.0698

Except for one case (for KIcS i w/b = 0.445, Table 4), the use of the F test for analysis of test results obtained within concrete with a constant water/binder w/b ratio indicated the difference between mean values of compressive strength and critical stress intensity factor of these concretes

The elimination of the results of the critical stress intensity factor obtained for concrete without additive and re-analysis of the mean equality indicates the lack of statistically significant differences between the mean values of the critical stress intensity factor KIcS of concretes modified with the selected additive (Table 4).

In the case of compressive strength test results only for the case w/b = 0.445 (Table 4), there is no reason to reject the hypothesis of equality of means.

Preceding further statistical analysis for all the results of the tested parameters covered by the research plan, the homogeneity of variance and significance of the impact of the adopted variables on the tested parameters were checked. In each case homogeneity of variance was demonstrated together with statistically significant (at the level of significance 0.05) influence of variables on the studied parameters.

Round  2

Reviewer 1 Report

Now ready for publication.